# The Impact of Air Source Heat Pump on the Production Performance of Broiler Chicks

Chenming Hu [1,2], Mohan Qiu [1,2], Chunlin Yu [1,2], Li Yang [1,2], Qubo Zhu [3], Anfang Liu [3], Longhuan Du [4] and Chaowu Yang [1,2,*]

1   Sichuan Animal Science Academy, Chengdu 610066, China
2   Animal Breeding and Genetics Key Laboratory of Sichuan Province, Chengdu 610066, China
3   College of Animal Science and Technology, Southwest University, Chongqin 400715, China
4   College of Architecture and Environment, Sichuan University, Chengdu 610017, China
*   Correspondence: chaowuyang@163.com

**Abstract:** Air source heat pump (ASHP) is a good new energy heating system. To explore the effect of ASHP on the production of yellow-feather broiler chicks, 31,500 one-day-old yellow broiler chicks were divided into three chicken houses with the same building structure but different heating methods (ASHP, CCF, CB). During the experiment, the parameters of heating time, temperature uniformity, gas concentration, weight gain, survival rate and production benefit were analyzed and evaluated. Results showed that the difference in $NH_3$, $CO_2$, and $H_2S$ concentrations was not significant in all test groups ($p > 0.05$). Only group II detected the CO gas. In winter and spring, the weight of the chickens in group II were weighed least at 35 days of age, and were significantly different from the ASHP and CB system ($p < 0.05$). There was no significant difference in body weight between ASHP and CB ($p > 0.05$). Group II had the lowest evenness and survival, the slowest warming, the worst uniformity of temperature distribution, and the highest cost. It is concluded that the ASHP was very environmentally friendly and has the highest economy, which is worth promoting and using.

**Keywords:** air source heat pump; yellow-feathered broiler chicken; chick rearing environment; production performance



## 1. Introduction

Coal was one of the most commonly used fuels in Southeast Asia and has been used as a heating fuel in China for thousands of years. In the breeding process of Chinese yellow feather broilers, coal boilers and flues, which were ancient heating methods, were still being used. Due to the needs of environmental protection and carbon neutrality, these ancient and extremely polluting heating methods must be eliminated. China had over 700 million farmers, most of whom were not large-scale breeding companies. Their main purpose in raising chickens was only to obtain the most basic economic income. However, during the energy revolution, they also had to upgrade and transform their heating methods. Unlike large companies, they did not have much wealth accumulation, so only heating methods that were both economical and suitable for China's national conditions were most suitable for them. For the upgrading of heating methods, large companies in China could use the most advanced methods internationally, but these methods were not suitable for most small- and medium-sized poultry breeders. In countries with developed animal husbandry, coal heating systems, especially honeycomb briquette heating systems, were almost not used in poultry production, so small- and medium-sized farmers had no international methods that could be used for reference too. In order to provide more references for farmers, the ASHP system has been included in the scope of this study.

In the processes of poultry farming (1–35 d), the factors that affect the breeding income were heating stability, chicken variety, and breeding environment. Generally speaking, the

highest productivity of chickens of the same breed was fixed, such as weight, egg production rate, etc. If high production efficiency was to be maintained continuously, it could only be achieved by improving heating conditions and the environment inside the chicken house. The Internal environment of poultry houses could be divided into gas environment and microbial environment [1,2]. Among them, the microbial environment mainly included pathogenic microorganisms and parasitic environments [3,4]. The disinfection technology in modern poultry breeding technology could effectively inhibit the reproduction of pathogenic microorganisms and parasites [1] but controlling the gas environment inside the breeding house remains a challenge [5]. The change of the temperature environment of the breeding house will affect the health of poultry [6], and then affect the commercial production and economic benefits in the later stage. In the cold areas of northern China, the heating period of chicken houses in winter can be more than 7 months, and in southern China, the heating time of chicken houses in winter will also reach more than 5 months.

With the rapid development of China's economy, the domestic demand for high-quality chicken has also further increased. By the end of 2021, the number of yellow-feathered broilers sold in China had reached nearly 5 billion. Compared with European and American countries, the development situation of animal husbandry in China is still relatively backward. Small- and medium-sized farmers do not have the economic strength to build new chicken farms, and many farmers still use traditional heating methods such as coal, gas, or electric heating. Although they want to improve the efficiency of equipment, they lack useful channels and experience. With the increase of the feeding density, the concentration of carbon dioxide, sulfur dioxide and ammonia gas in the house will also rise sharply [7]. These gases not only have an impact on the health of farmers, but also greatly damage the welfare of animals [8,9].

With increasing environmental protection [10] and more attention being paid to the environmental welfare of animal feeding in China [11,12], traditional heating methods are in urgent need of replacement by a more effective, more economical, more environmentally friendly and more conducive microclimate regulation system. Although the ground source heat pump system [13,14] and solar heat pump also belongs to this aspect, their installation and procurement costs are too high for Chinese farmers and the construction is also difficult [15,16]. In China, the power resources are relatively rich, and the ASHP has been fully used in residential housing because the installation method is simpler, such as for hotel heating, water supply, office cooling, energy conservation and so on [17,18]. The living environment has quite strict control of harmful gases, so it is potentially feasible to install the air source heat pump in the poultry house for heating and improve the air quality and welfare of chicks in the house.

The ASHP is composed of an outdoor unit and a thermal insulation water tank. The outdoor unit is called the air-energy heat pump unit. The ASHP unit is generally composed of an expansion valve, compressor, evaporator, condenser and other main components. Its working principle is that cryogenic refrigerant in the compressor system continuously absorbs the low heat in the outdoor air and then brings it back to the compressor to heat the cold water. The air source heat pump also has excellent performance of working at high temperature and high energy conversion efficiency [19].

As a system equipment which effectively collects and transfers heat, the ASHP can convert the power consumed into three or five times the heat, to achieve the purpose of high efficiency, energy saving, and low carbon emissions. In order to better optimize the heating facilities of small- and medium-sized farmers in China and solve the existing problems of high energy consumption and bad air in their chicken houses, we compared the ASHP, CCF and CB system along multiple parameters. The economy of ASHP and its impact on animal welfare were discussed to provide a new reference method for the production improvement of yellow-feathered broilers in China.

## 2. Materials and Methods

### 2.1. Heating System Design

2.1.1. ASHP Brief Description

In November 2017, an ASHP heating system was built in a professional poultry company's broiler house in Sichuan Province, China. The system contains an air can pump host, a stainless-steel insulation bucket, a two pipe circulating pump, a series of stainless-steel pipes connected to the chicken house and ten indoor radiator (Figure 1), and a series structure link was adopted between the radiators [20]. Through the compressor work of the ASHP, the water temperature can be raised to 50 to 60 °C from the room temperature. The temperature inside the chicken house was regulated by an automatic temperature control module. The energy efficiency calculation method for ASHP heating water was as follows.

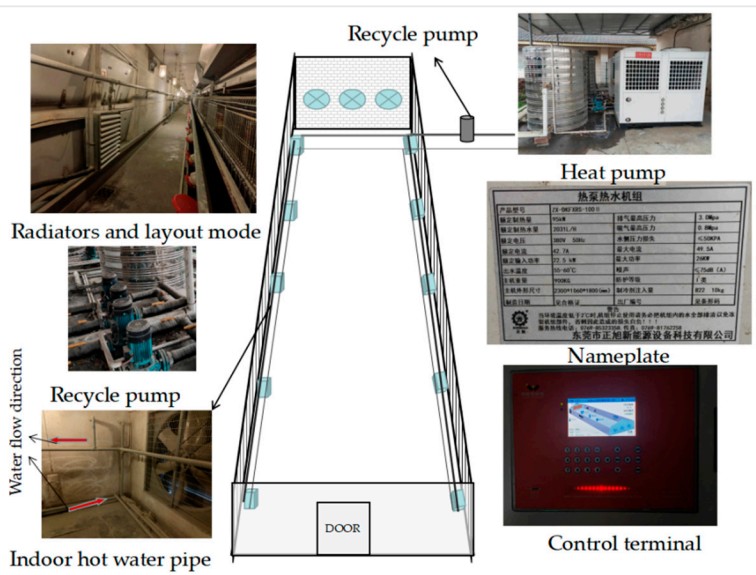

**Figure 1.** Schematic diagram of the ASHP heating system.

The formula for the total energy required to heat 1 ton of water to the target temperature using ordinary electric heating was as follows:

$$Q1 = m \cdot c \cdot (t1 - t2) \div W1 \tag{1}$$

**Formula 1.** Energy consumption formula.

Q1: Total energy required (KWh), m: The weight of water (ton), c: Specific heat capacity of water (MJ/ton·°C), t1: Target water temperature (°C), t2: Initial water temperature (°C), W1: Heat provided by 1 unit of energy (MJ).

Due to the highest utilization rate of electricity being only 52%, the actual power consumption formula was as follows:

$$Q2 = Q1 \div 0.52 \tag{2}$$

Q2: Actual power consumption (KWh).

The energy efficiency ratio formula for ASHP:

$$Q3 = \sigma \div W2 \tag{3}$$

Q3: Energy efficiency ratio, σ: Rated power of ASHP (KWh), W2: Input power (KWh).

The formula for the electrical energy required for ASHP to raise 1 ton of water by 1 °C was as follows:

$$Q4 = Q2 \div Q3 \tag{4}$$

Q4: Total electricity required (KWh).

### 2.1.2. Cellular Coal Flue Brief Description

At the same company which installed the ASHP chicken house, there was also a cellular coal flue system (CCF) consisting of ten additional points and a number of connected flues (Figures 2 and 3). The layout position of the honeycomb coal adding point was the same as that of the radiator. CCF controls the heat source temperature by increasing or decreasing the amount of honeycomb coal added. In order to avoid the harm of sulfur dioxide to the experimenter, we used desulfurization honeycomb coal as fuel [21].

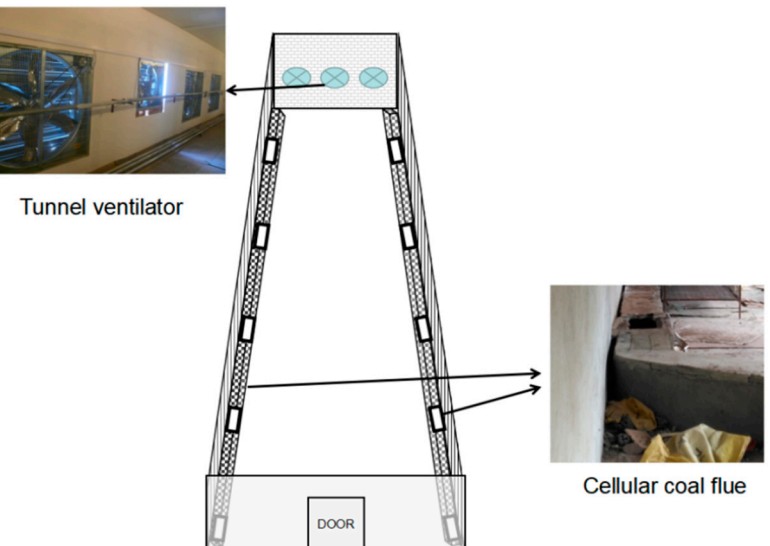

**Figure 2.** Schematic diagram of the CCF system.

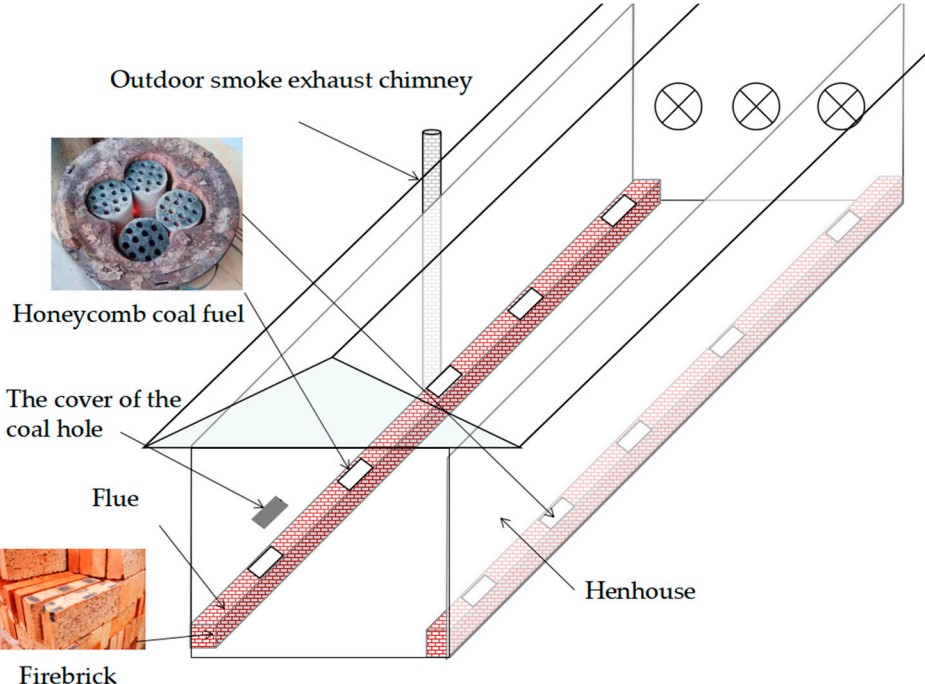

**Figure 3.** Schematic diagram of the flue installation layout.

The main body of the CCF flue was constructed of firebrick (Figure 4). Farmers rely on adding honeycomb coal to the flue to raise the temperature of the flue and use heat transfer to raise the temperature in the chicken house. Honeycomb coal was a large honeycomb

coal block that is burned as a fuel in the honeycomb coal flue and was the main household fuel for many inhabitants of East Asia. The main components of honeycomb coal were raw coal, carbonized sawdust, lime, red (yellow) mud, charcoal powder and other mixed materials and nitrate, potassium permanganate, etc. (Figure 4).

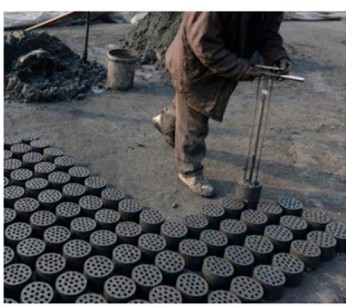
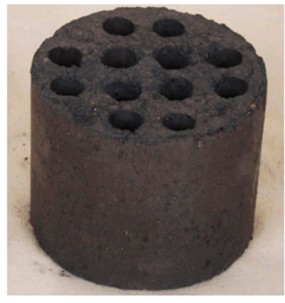

Handmade honeycomb coal

Finished honeycomb coal fuel
(Before combustion)

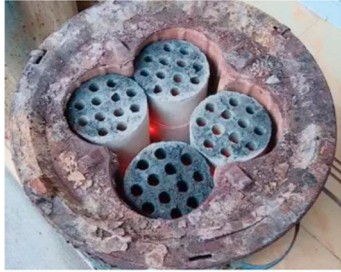
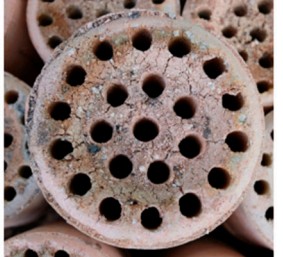

Honeycomb coal fuel
(Under combustion in special stove)

Honeycomb coal fuel
(After combustion)

**Figure 4.** Fuel honeycomb coal for CCF system.

The CCF system did not have an automatic coal feeding function. Farmers needed to conduct a simple combustion test before using honeycomb coal to determine the combustible time of this batch of coal. When officially used, farmers determined the replacement time of honeycomb coal based on the previously measured combustible time. In this experiment, each honeycomb coal hole could accommodate 50 pieces of honeycomb coal at once, and the replacement interval of honeycomb coal was once every 2 h. When replacing honeycomb coal, one must first remove each piece that had been completely burned, and then add the same amount of new coal as the amount that has been removed.

The formula for energy consumption can be found in Formula 1.

Because honeycomb coal was mainly composed of about 70% peat and about 30% soil, its unit heat could only reach up to 70% of standard coal.

$$Q2 = Q1 \div 0.7 \tag{5}$$

Q2: Total Honeycomb coal required (ton).

### 2.1.3. Coal-Fired Boilers Brief Description

At the same company that installed the two appeal heating systems, there was also a coal-fired boilers (CB) system. The system consists of a coal-fired boiler, a pipe circulating pump, a smoke exhaust machine, a series of stainless-steel pipes connected to the chick house and ten indoor radiators (Figure 5). The temperature inside the chicken house was adjusted by a temperature regulator. CB controls water temperature by increasing or less addition frequency and amount of coal.

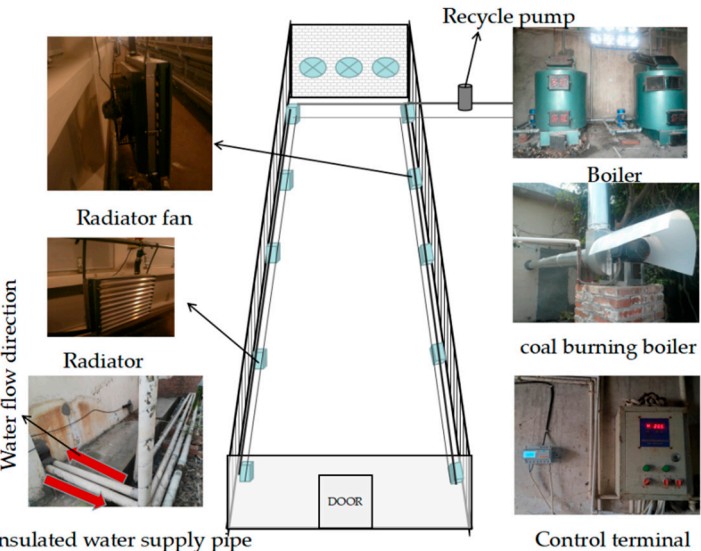

**Figure 5.** Schematic diagram of CB system.

The formula for energy consumption can be found in Formula 1.

### 2.1.4. Broiler and Experimental Design

The test site was a professional, yellow-feathered broiler breeding company, and the parameters were compared in winter and spring. On the day of the winter test, the outdoor temperature was 9 to 13 °C, and on the day of the spring test, the outdoor temperature was 10 to 21 °C. Because of the high temperature in southern China, summer and autumn were not included in the trial design. This experiment used three equally constructed environmentally controlled commercial broiler houses, all measuring 10 m wide, 40 m long, and 3 m high. The house is installed with a welfare chicken cage, each cage was 2 m long, 1.4 m wide and 0.6 m high, and a total of 144 such cages were installed in each chicken house. One of the chicken houses installed with ASHP had a capacity of 95.9 kW (Zhengxu New Energy Equipment Technology Co., LTD., Dongguan, China), and the other chicken house was CCF, and the remaining one was the CB system.

In total, 31,500 one-day-old chicks were randomly divided into three large groups of 10,500 in each. Three replicates were set within each large group, with 3500 chickens each. The testing was performed for a total of 35 days. During the test, all flocks ate the same ingredient diet (Table 1) and drank the same water. Feed was supplied in unlimited quantities every day. The requirements for rearing temperature were as follows: 36 °C (1 to 3 d), 32 °C (4 to 7 d), 30 °C (8 to 15 d), and 25 °C (16 to 35 d). All three experimental treatments were timed to monitor the temperature, RH, $CO_2$, CO, $H_2S$ and $NH_3$. During the test, all the upper limits of hazardous gas concentrations met the requirements for animal welfare.

### 2.2. Measurement Method

### 2.2.1. Production Performance

During the experiment, chicks from the three test houses were weighed weekly until the 35th day of age and the mortality rate of broilers was calculated weekly and recorded as a percentage. On the last day (35th day) of the trial, 10% samples were taken from each flock for weight weighing and the end weight and evenness of the flock was compared.

### 2.2.2. $CO_2$, CO, $H_2S$, $NH_3$ Gas Concentration and Temperature, RH

Gases such as $CO_2$, CO, $H_2S$, and $NH_3$ are the main factors affecting the environment inside enclosed chicken coops, with high concentrations that are easy to detect. In particular, $NH_3$ and CO are the main causes of respiratory diseases in chickens. Excessive $CO_2$ concentration can cause a decrease in oxygen content in the chicken coop, while also

causing a local greenhouse effect, increasing the heat dissipation power consumption of the coop and reducing the feed intake of the chicks. By evaluating the concentrations of these gases, the environment inside the chicken coop can be effectively evaluated.

**Table 1.** A standard corn-soybean meal based diet formula of broiler chickens.

| Composition | Content |
|---|---|
| Corn (%) | 57.26 |
| Soybean meal (%) | 23.61 |
| Corn protein powder (%) | 6.0 |
| Corn germ meal (%) | 8.0 |
| Salt (%) | 0.30 |
| Mountain flour (%) | 1.33 |
| Calcium hydrogen phosphate (%) | 1.9 |
| Lysine (%) | 0.34 |
| Methionine (%) | 0.16 |
| Soya-bean oil (%) | 0.8 |
| Antigen King (%) | 0.02 |
| Mineral Additives (%) | 0.15 |
| Anti-mildew agent (%) | 0.06 |
| Multivitamins for poultry (%) | 0.05 |
| Phytase (%) | 0.02 |
| Poultry metabolic energy (Kcal/kg) | 2.90 |
| Crude protein (%) | 21.00 |
| Calcium (%) | 1.00 |
| Total phosphorus (%) | 0.72 |
| Non-phytic acid phosphor (%) | 0.43 |
| Na (%) | 0.14 |
| Lysine (%) | 1.04 |
| Methionine (%) | 0.48 |

Parameter measurement: The multifunctional composite gas analyzer was used to measure the concentration of $NH_3$, $H_2S$ and $CO_2$ (model GT2000, Corno Electronic Technology Co., LTD., Dongguan, China), with a range of 0 to 10,000 ppm and the detection accuracy is ±3%. CO was measured using a CO rapid detector (model ADKS-1 from Edex, Changzhou, China), with a range of 0 to 1000 ppm and the detection accuracy is ±2%. The temperature and humidity digital tester was used to determine the inner temperature and humidity (model ASB817, Wanchuang Electronic Products Co., LTD., China), the temperature measurement range was from −10 to 50 °C (±3%), and the humidity measurement range was 5 to 98% (±3%).

A total of six points were selected for each nursery to determine the temperature and the concentration of $NH_3$, $H_2S$, CO, and $CO_2$. The measurement points are numbered as shown in Figure 6, and the measurement height was 50 cm. Points i and iii were the heat sources. Point iv was the cage measurement point which closest to the heat source on the left side of the coop. Points ii, v, and vi were the longitudinal cutting point in the middle of the chicken house, the distance between the points was 15 m, 50 cm high from the ground. When measuring the temperature in the test, we first determined the temperature value of points ii, iv, v and vi, respectively, and then averaged the value of these temperatures to be a new temperature data and regarded the new temperature data as the actual arrival temperature of the chicken house. Parameter measurements were performed at 1 pm on trial days 3, 7, 15 and 35. It was particularly noteworthy that due to the high outdoor environmental temperature in spring, the ventilation system of the chicken house was opened too often, which had an adverse impact on the gas measurement. In order to more accurately judge the law of gas concentration value, we only did the gas concentration measurement in winter, not in spring [22]. The layout of the chicken coop in the henhouse was shown in Figure 7.

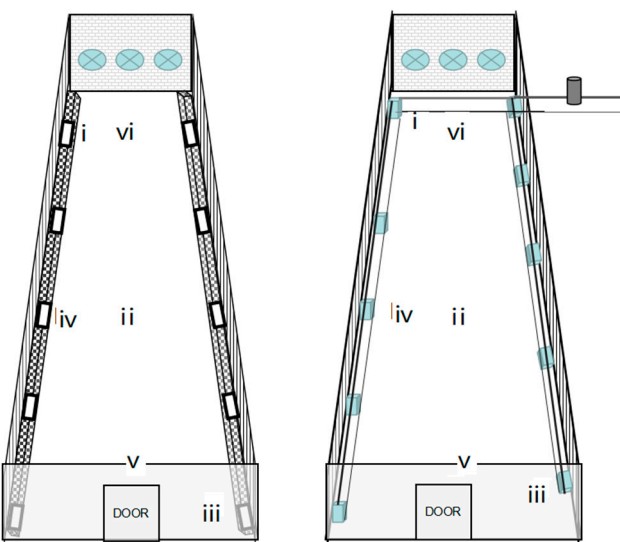

**Figure 6.** Schematic diagram of the parameter measurement points. i, iii and iv: Measurement points close to the heat sources, v, ii and vi: Measurement points at the front, middle, and back of the central chicken coop.

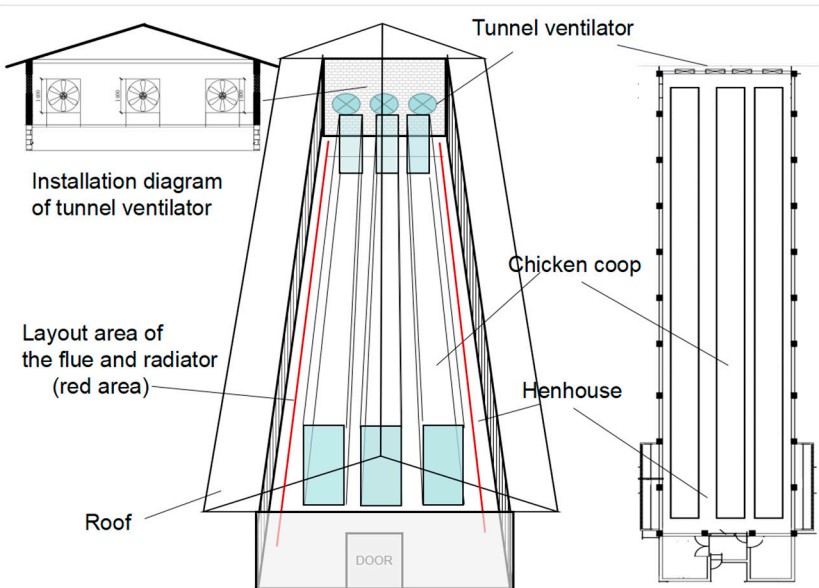

**Figure 7.** Schematic diagram of the chicken cage layout in the chicken house.

Heating rate: On the day of the winter test, the temperature outside the house was 7 to 11 °C, and the measurement point iv was 13 °C. The time consumed to increase the temperature at the measurement point iv from 13 to 25 °C and from 13 to 36 °C was used as a comparative indicator of the heating efficiency of the winter chicken house. On the day of the spring test, the outside temperature was 9 to 20 °C, and the temperature of measurement point iv was 14 °C. The time consumed to increase the temperature of measurement point iv from 14 to 25 °C and from 14 to 36 °C was used as the comparative index of the spring heating speed.

### 2.2.3. Estimation of Energy Consumption and Cost

ASHP power consumption was calculated based on the total reading of the meter. CCF was calculated based on the total recorded honeycomb coal consumed during the trial period. CB was based on the weight of the total purchased coal at the beginning of the test minus the weight of the remaining coal at the end of the test. Labor cost and

equipment purchase cost are calculated according to actual expenditure. The theoretical life of equipment was the average life of similar equipment. During the experiment, the calculation of the test comprehensive cost was carried out in RMB.

### 2.3. Data Statistics and Analysis

Initial trial data were first preprocessed with the excel software. The comprehensive data results were subjected by a paired samples T-test using SPSS 10.0. Data from all measured items are expressed as the mean, plus or minus the standard deviation. $p < 0.01$ was indicated when the difference was extremely significant and $p < 0.05$ when significant.

## 3. Results

### 3.1. Time Consumed for Heating

Temperature was an important guarantee for the healthy growth of chickens. The rate of temperature rises in different experimental groups as shown in Table 2, CCF was the slowest, CB was faster than the CCF group, and ASHP was the fastest. The average temperature of measuring points ii, iv, v and vi was taken as the actual temperature of the chicks in the house. The absolute value of the difference between the actual temperature value of the chicks and the actual temperature standard value of the chicks was the deviation of the temperature of the chicks, and the larger the deviation value, the worse the temperature uniformity in the house (Tables 3 and 4).

**Table 2.** Comparison of heating efficiency for different test groups.

| Group | Winter | | Spring | |
|---|---|---|---|---|
| | **13 to 25 °C** | **13 to 36 °C** | **14 to 25 °C** | **14 to 36 °C** |
| I (h) | 3.8 | 8.7 | 2.8 | 6.0 |
| II (h) | 3.1 | 4.5 | 2.3 | 3.7 |
| III (h) | 1.6 | 2.5 | 0.8 | 1.5 |

Heating efficiency = Stop heating time–Start heating time, I = air source heat pump (ASHP), II = cellular coal flue (CCF), III = coal-fired boilers (CB).

**Table 3.** Comparison of the temperature and RH in the chicken house in winter (1–35 d).

| Age (d) | Standard Temperature (°C) | Mean (°C) | | | RH (%) | | | Temperature Deviation Value [1] (°C) | | |
|---|---|---|---|---|---|---|---|---|---|---|
| | | **I** | **II** | **III** | **I** | **II** | **III** | **I** | **II** | **III** |
| 3 | 36.0 | 36.4 | 33.6 | 36.2 | 57.8 | 65.8 | 58.5 | 0.4 | 2.4 | 0.2 |
| 7 | 32.0 | 32.5 | 30.2 | 32.6 | 62.2 | 63.3 | 57.5 | 0.5 | 1.8 | 0.6 |
| 15 | 30.0 | 29.8 | 28.0 | 29.8 | 56.5 | 65.5 | 55.9 | 0.2 | 2.0 | 0.2 |
| 35 | 25.0 | 25.1 | 24.3 | 25.1 | 58.3 | 60.5 | 53.5 | 0.1 | 0.7 | 0.1 |
| Mean ± SD | - | - | - | - | - | - | - | 0.3 ± 0.2 | 1.7 ± 0.7 | 0.3 ± 0.2 |

[1] Temperature deviation value = |Mean–Standard temperature|, I = air source heat pump (ASHP), II = cellular coal flue (CCF), III = coal-fired boilers (CB).

**Table 4.** Comparison of the temperature and RH in the chicken house in spring (1–35 d).

| Age (d) | Standard Temperature (°C) | Mean (°C) | | | RH (%) | | | Temperature Deviation Value [1] (°C) | | |
|---|---|---|---|---|---|---|---|---|---|---|
| | | **I** | **II** | **III** | **I** | **II** | **III** | **I** | **II** | **III** |
| 3 | 36.0 | 35.9 | 35.8 | 35.3 | 58.8 | 67.5 | 57.5 | 0.1 | 0.8 | 0.2 |
| 7 | 32.0 | 32.1 | 32.4 | 33.5 | 58.5 | 67.3 | 52.7 | 0.1 | 1.5 | 0.4 |
| 15 | 30.0 | 29.9 | 30.1 | 31.2 | 60.2 | 65.8 | 55.5 | 0.1 | 1.2 | 0.1 |
| 35 | 25.0 | 25.7 | 25.9 | 23.9 | 52.3 | 62.5 | 60.2 | 0.7 | 1.1 | 0.9 |
| Mean ± SD | - | - | - | - | - | - | - | 0.3 ± 0.3 | 1.2 ± 0.3 | 0.4 ± 0.4 |

[1] Temperature deviation value = |Mean–Standard temperature|, I = air source heat pump (ASHP), II = cellular coal flue (CCF), III = coal-fired boilers (CB).

In winter, the temperature deviation value in the CCF system was too large, the temperature uniformity was poor, the temperature uniformity of ASHP and CB was good.

These conditions reappeared in the spring trial. The reason may be that the CCF had a high requirement for fuel replacement frequency, especially at night. If the fuel was not replaced in time, the temperature in the chicken house would fluctuate greatly in a short time.

### 3.2. Main Environmental Factors

Generally speaking, the factors that affect the residential environment mainly include indoor air quality, temperature, humidity and harmful gas concentration [23,24]. The humidity value in the test chicken houses with three different heating systems was between 50–65% (Table 5), so it could be seen that the humidity environment of the three chicken houses was relatively ideal.

**Table 5.** Effect of different heating systems on $NH_3$, $H_2S$, CO, and $CO_2$ gas emissions in chicken houses during the experimental period (1–35 d).

| Age (d) | | $NH_3$ Content (ppm) | | | $H_2S$ Content (ppm) | | | CO Content (ppm) | | | $CO_2$ Content (ppm) | |
|---|---|---|---|---|---|---|---|---|---|---|---|---|
| | I | II | III | I | II | III | I | II | III | I | II | III |
| 3 | 2.5 | 2.3 | 2.4 | 0.3 | 0.2 | 0.2 | 0 | 22.5 | 0 | 5035.3 | 6931.4 | 5098.0 |
| 7 | 2.0 | 1.8 | 1.8 | 0.3 | 0.3 | 0.2 | 0 | 17.5 | 0 | 4727.5 | 5860.8 | 4666.7 |
| 15 | 1.8 | 1.8 | 1.7 | 0.2 | 0.2 | 0.1 | 0 | 10.0 | 0 | 4156.9 | 4294.1 | 4411.8 |
| 35 | 1.4 | 1.5 | 1.4 | 0.2 | 0.2 | 0.2 | 0 | 3.8 | 0 | 3721.6 | 3823.5 | 3764.7 |
| Mean ± SD | 1.9 ± 0.5 | 1.9 ± 0.3 | 1.8 ± 0.4 | 0.3 ± 0.1 | 0.2 ± 0.05 | 0.2 ± 0.9 | 0 | 13.5 ± 8.2 | 0 | 4110.3 ± 585.9 | 5227.4 ± 1431.4 | 4485.3 ± 557.7 |

| | | $NH_3$ | | | $H_2S$ | | | CO | | | $CO_2$ | |
|---|---|---|---|---|---|---|---|---|---|---|---|---|
| *p*-value | I-II | 0.61 | | | 0.39 | | | 0.04 | | | 0.16 | |
| | II-III | 0.64 | | | 0.51 | | | 0.04 | | | 0.21 | |
| | I-III | 0.09 | | | 0.38 | | | - | | | 0.34 | |

I = ASHP (air source heat pump), II = CCF (cellular coal flue), III = CB (coal-fired boilers).

$NH_3$, $H_2S$, and $CO_2$ were the most common gas in closed chicken houses, and the gas emissions were higher in winter than in spring [25]. The ambient gas measurement values during the test were shown in Table 4. The concentration difference of $NH_3$, $H_2S$, and $CO_2$ in all test groups was not significant ($p > 0.05$). CO was detected in the group CCF, but not in the ASHP and CB groups. The CO concentration of CCF was significantly different between ASHP and CB ($p < 0.05$).

### 3.3. Production Performance

There were many factors affecting the production performance of chicks, such as genetics and nutrition. With the same breed, coop structure and feed, the difference in production level was greatly related to the microclimate environment in the house [26]. Chicks in the CCF group had the lightest weight at 35 days of age (Figure 8) of all experimental groups and had the highest mortality rate (Figure 9). Body weight and mortality in the CCF group varied significantly from the ASHP and CB groups ($p < 0.05$). Especially for mortality, the CCF group already exceeded the sum of ASHP and CB. It may be due to the poor temperature uniformity and the gas environment in the CCF chicken house.

### 3.4. Capitalized Cost

In the heating work of chick houses, the highest costs are the labor cost and fuel cost. In the present experiment (Table 6), the ASHP system did not consume any stone fuel in operation [27]. So, the cost of the ASHP group was lower than that of the CCF group with the highest cost, and the heating cost of ASHP was only about 50% of that of CCF. At the same time, both the CCF and CB system require two people to be on duty 24 h in turn, which also violates human welfare requirements. The operation of ASHP was intelligent. When the system fails, the alarm in the machine would automatically start working to remind the management personnel of the mechanical failure of the system, so the equipment did not need the farmer to be on duty all the time. The workload of the workers was very small. Ordinary farmers can be competent for the work without paying more wages, so the labor cost was also very low. Although the ASHP system used the most

expensive equipment, it had the highest heating efficiency and the lowest energy and labor costs, so the combined operating costs were the lowest.

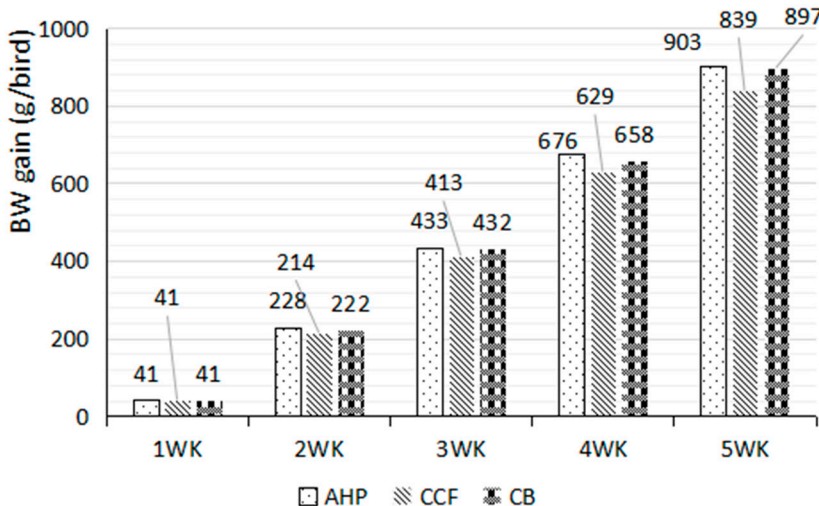

**Figure 8.** Comparison of broiler body weight values at 35 d of age in different heating system groups. (ASHP and CCF: $p < 0.05$, ASHP and CB: $p > 0.05$, CCF and CB: $p < 0.05$, n = 3500), ASHP = air source heat pump, CCF = cellular coal flue, CB = coal-fired boilers.

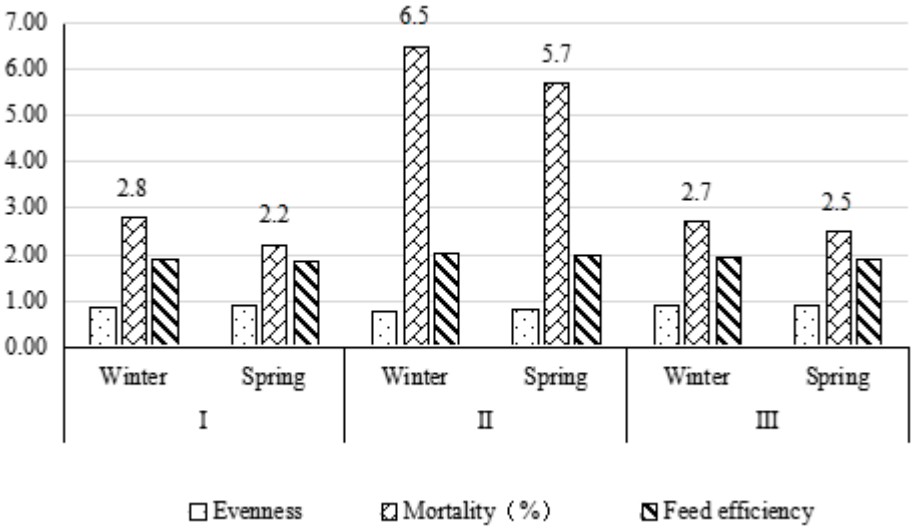

**Figure 9.** Effect of heating system on the evenness, mortality, and feed efficiency of chickens during the experimental period (1–35 d), I = air source heat pump (ASHP), II = cellular coal flue (CCF), III = coal-fired boilers (CB).

**Table 6.** Comparison of total energy consumption and total operating costs during the three chicken houses (1–35 d).

|  |  | I | | II | | III | |
|---|---|---|---|---|---|---|---|
|  |  | **Spring** | **Winter** | **Spring** | **Winter** | **Spring** | **Winter** |
| Electricity | Consumption (kWh) | 1320 | 3816 | - | - | - | - |
|  | Unit-price (yuan) | 0.6 | 0.6 | - | - | - | - |
| Honeycomb coal | Consumption (ton) | - | - | 8.7 | 15.2 | - | - |
|  | Unit-price (yuan) | - | - | 720 | 720 | - | - |
| Coal | Consumption (ton) | - | - | - | - | 6.2 | 9.7 |
|  | Unit-price (yuan) | - | - | - | - | 800 | 800 |

**Table 6.** *Cont.*

|  | I | | II | | III | |
|---|---|---|---|---|---|---|
|  | **Spring** | **Winter** | **Spring** | **Winter** | **Spring** | **Winter** |
| Fuel cost (yuan) | 792 | 2290 | 6264 | 10,944 | 4960 | 7760 |
| Labor cost (yuan) | 4000 | 4000 | 4500 | 4500 | 4500 | 4500 |
| Equipment cost (yuan) | 528 | 528 | 104 | 104 | 321 | 321 |
| Composite cost (yuan) | 5320 | 6818 | 10,868 | 15,548 | 9781 | 12,581 |
| Mean ± SD (Composite cost) | 6069 ± 1059 | | 13,208 ± 3309 | | 11,181 ± 1980 | |
| Chicken income (yuan) [1] | 265,188 | | 246,372 | | 263,777 | |
| Economic benefits (yuan) [2] | 259,119 | | 233,164 | | 252,596 | |
| *p*-value (Composite cost) I–II | | | 0.04 | | | |
| II–III | | | 0.02 | | | |
| I–III | | | 0.02 | | | |

[1] $n = 21{,}000$, $n$ (Spring) = 10,500, $n$ (Winter) = 10,500. [2] Economic benefits = Chicken income–Composite cost.

One US dollar = 6.8 yuan (as of February 2023), honeycomb coal price = 720 yuan/ton, electricity price = 0.6 yuan/kWh, and coal price = 800 yuan/ton (as of February 2023). I = air source heat pump (ASHP), II = cellular coal flue (CCF), III = coal-fired boilers (CB).

## 4. Discussion

From the perspective of the development trend of animal husbandry in China, to save costs, increase efficiency and improve the breeding environment has become an important goal. This study provided a reference heating method for small- and medium-sized farmers in China, and also proved that the ASHP system can effectively replace CCF and CB.

In the production of broilers, the biomimetic environment has always been simulated, just like searching for the optimal temperature range for chick growth. It has been reported that 33 °C would not affect the health of chicks and could save energy [28]. However, since the temperature of hens was about 36 °C, the standard temperature of 35 to 36 °C for within 3 days of age could effectively improve the welfare of broilers [29]. ASHP had a more stable temperature supply and was the most efficient welfare system of the three heating methods.

The ASHP, CCF and CB heating systems had three different management and operation modes. In the terms of management, CCF belonged to the fully manual intervention type, and all operations required manpower to complete. The stability of the system heating could only rely on the farmers' own management ability, but only very few farmers could meet the production requirements. When the CCF system was started, farmers needed to use honeycomb coal to raise the temperature of the flue up to 60 to 80 °C and they needed to replace the fuel at regular intervals later to maintain enough heat in the flue. CB belonged to a semi-automatic system, where administrators only needed to add coal, and indoor temperature control was handed over to temperature sensors for control. The dependence of the CB system on farmers' management level had decreased. ASHP was the most advanced system. As long as the breeding company did not cut off power or water, the system could operate intelligently 24/7 under the control of preset parameters with almost no delay and completely liberated the workforce.

Different from the ASHP and CB, the temperature of the CCF had been fluctuating, which had had some impact on the growth of chicks. During the course of the study, the chickens in the CCF group also suffered from some diseases, such as coli infection and respiratory tract infection. In order to make the test results more accurate, the CCF group was not given additional therapeutic drugs, which may also be one of the reasons for the highest mortality rate in the CCF group. In actual production, when the chickens were infected with bacteria, farmers would immediately feed them with therapeutic drugs to reduce the mortality, especially with the use of antibiotics which could affect chicken sales, whenever chickens develop bacterial infections [30]. Due to the lower temperature in the

early stage of the CCF group, the chicken herd also experienced crowding from the first to third days, resulting in a corresponding decrease in feed intake. The poor temperature environment caused the chicks to miss the first peak period of growth (1–3 d), resulting in higher feed meat in the CCF group and the lightest final body weight of the chicks. The occurrence of respiratory diseases may be closely related to the presence of CO in the environment [31].

One of the aims of this study was to replace coal fuels. Coal is not an environmentally friendly fuel, its combustion produces a large number of greenhouse gases, such as $CO_2$, and incomplete combustion would produce CO. $CO_2$ had a huge impact on the environment, which was why China changed its strategy to reduce carbon emissions from animal husbandry [12]. In the CCF system, the burning point of the honeycomb coal was in the chicken house. The burning of honeycomb coal consumes oxygen and releases large amounts of $CO_2$, which reduced inside air quality [32]. Although most of the $CO_2$ would be discharged outside the house through the smoke vent, the flue was not a completely sealed space, and some $CO_2$ would still enter the chicken house. Incomplete combustion of coal also produces CO, which was the main reason for detecting CO in the CCF group. The coal outlet of the coal boiler was outside the house, so the $CO_2$ emissions within the chicken house in the CB system was close to ASHP, both lower than the CCF, and the CO gas was not detected. The small- and medium-sized chicken farms in China were not very modern, and the age of chicken farmers was also relatively old. Not only did CO have a huge impact on the health and growth environment of poultry [31], but it also caused great harm to the health of farmers [33,34]. When Zhang et al. [35] and Wei Wan et al. [36] studied honeycomb coal, it was found that honeycomb coal also contained heavy metal lead and aromatic compounds. This was one of the important reasons why the CCF and CB systems must be eliminated as soon as possible.

Although it is inevitable that coal will be used for many years, China has already tried to reduce its carbon emissions. In order to achieve this goal, China has gradually reduced coal mining production of coal products. Both honeycomb coal and coal belong to the coal products. Small honeycomb briquette workshops and coal mining enterprises were also gradually closed because they could not meet the environmental protection requirements, and this has led to a decline in coal supply and a sharp increase in prices. On the other hand, small honeycomb coal workshops and coal mining companies were both non-standard producers, and their formulas were not fixed, which also leads to unstable fuel quality and affects farmers' profits.

## 5. Conclusions

This study investigated three heating systems: ASHP, CB and CCF. These three systems were all installed in the same structure of the chicken coop. In order to compare the differences and advantages of the three systems, heating efficiency, usage cost, impact on chicken production, and environment were all included in the evaluation scope. The research results indicate that it was feasible for small- and medium-sized farmers in China to upgrade the heating system of chicken houses using ASHP heating systems.

The ASHP was very friendly to the environment of the chicken house and animal welfare. It can not only improve the microclimate environment of the farm, reduce CO and carbon dioxide emissions, improve the production efficiency and economic benefits of broiler breeding, but it also liberates the labor force of farmers. Compared with the other two traditional heating systems, the comprehensive performance of ASHP makes it more promising for further promotion in the chicken industry.

**Author Contributions:** Conceptualization, C.H. and C.Y. (Chaowu Yang); methodology, C.H.; software, C.Y. (Chunlin Yu); validation, M.Q., Q.Z. and L.Y.; formal analysis, A.L.; investigation, C.H.; resources, C.Y. (Chaowu Yang); data curation, L.D.; writing—original draft preparation, C.H.; writing—review & editing, C.Y. (Chaowu Yang); visualization, C.Y. (Chunlin Yu); supervision, M.Q.; project administration, L.Y. All authors have read and agreed to the published version of the manuscript.

**Funding:** This work was funded by Sichuan Province key research and development plan (2020YFN0146, 2021YFN0029, 2021YFYZ0031); Sichuan Provincial financial operation special project (SASA2020CZYX002); China Agriculture Research System (grant number CARS-41-G07).

**Data Availability Statement:** The data used in this paper is only a part of the entire project dataset. Due to privacy reasons, data is not suitable for public disclosure.

**Conflicts of Interest:** The authors declare no conflict of interest.

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
