# Peer review of "The Impact of Air Source Heat Pump on the Production Performance of Broiler Chicks"

_processes, doi:10.3390/pr11051360_

Round 1
Reviewer 1 Report
Dear authors:
It is necessary to add a table of uncertainties of the equipment’s used, in order to strengthen the safety of the measurements.
The authors comment that the weights of the chickens were recorded, however, there is no table or graph to observe this process.
A paragraph should be added to explain the importance of the variables to be measured (CO2, CO, H2S, NH3 Concentration and gas temperature, HR)
Author Response
Thank you for your modifications and suggestions. I have rearranged and annotated the article according to the formatting requirements. I hope it meets the requirements of the magazine. Thank you. Best wishes. The previous material had an error and has been replaced.

Reviewer 2 Report
The heat pump is a very efficient technique for the application of temperature rise that is not too high because of a very high COP. This article still contains a lot of writing errors, both in terms of punctuation and the correct way of citation. To improve the quality of this article, the following should be corrected:
1. Use “hyphen” and “en dash” correctly. Many mistakes or confused the use of the two signs.
2. The referral technique to the Bibliography uses the number according to the style used by this journal.
3. Measurement method should be added.
4. Main formulation and method for evaluating should be clearly written.
5. Conclusion of this research must be added after discussion.
6. What is the meaning of Mean+SD in Table 5?
7. More detailed comments can be seen in the attached draft.

Author Response
Thank you for your suggestion and support. I have made adjustments to the format and layout of the article, and made corresponding additions to the discussion section. thank you. Best wishes. The previous material had an error and has been replaced

Reviewer 3 Report
The review of the manuscript entitled “The impact of air source heat pump on the production performance of broiler chicks.”
The authors made experimental research into comparison between three different heat generators of the chicken coops. They researched an impact of the heat generator on the indoor air quality in the room where the chickens are bred, which is the immediate outcome. Eventually they analyzed the final results: weight gain, mortality, and income and expenditure after 35-day breeding.
The paper needs some minor improvements.
The description of the Cellular Coal Flue system in the subsection 2.1.2. is too briefly. It should be described in detail, for it is not known as widely as ASHP, or a coal boiler system. It ought to be sketched the heat generator(s), duct system, and CCF control devices. The last are mentioned in the line 93.
What material is CCF made from? Is the honeycomb coal either a substance for CCF construction or burnt fuel? What fuel is burnt?
Figure 2. is ambiguous. Where is the tunnel ventilator? Is the room in the figure centre either the chicken coop or the tunnel ventilator (ventilation duct)?
Line 143. Where is the cage located? It should be sketched in the diagrams.
All the control devices mentioned in the lines 193-194 should be sketched in Figures 1, and 3.
Please write down the formula for heating efficiency calculation whose Table 1 shows.
Please write down the formula for P-value in Tables 2 and the next one.
Why I-III is 0.72? If I-II and II-III are 0.02, shouldn't be I-III= 0.04? Appropriately please explain these differences in Tables 3, 4, and 5.
Table 5. I assume the composite costs are the sum of Labour, equipment, and fuel or power costs. However there is no fuel costs, the authors should calculate them.
Since the section 3 is quite long I would advise to add the section 4 where the most important conclusions would be enumerated briefly.
Author Response

(The authors gave the same response as above.)

Round 2
Reviewer 2 Report
Some important things that must be corrected are:
1. Conclusion must be separated in a section---> 5. Conclusion
2. The use of punctuation in references needs to be improved.
3. Other comments please see the draft (enclosed)

Author Response
Thank you again for your guidance. The reference format of the paper has been modified according to the suggestions in the PDF. The fifth section has been added to the paper. I hope the new manuscript meets the requirements.

Reviewer 3 Report
The review of the manuscript entitled “The impact of air source heat pump on the production performance of broiler chicks.”
The authors made almost all the corrections, for I couldn’t find the formulas for P-values. Additionally the corrected part needs further minor improvements.
Lines 113-116. There are swapped symbols of specific heat and mass. Specific heat should be denoted with small letter "c", but small letter "m" is symbol of mass. By a common agreement the symbol of temperature in Celsius scale is small letter "t", while temperature in Kelvins is written down with "T". The multiplication symbol is "·"- left Alt + 0183 on the numeric keyboard.
Lines 135-136. The honeycomb coal must be added manually (cf. lines 364-367), so how does CCF control the amount of added fuel?
Lines 147-150 cf. lines 113-116.
Lines 173-175 cf. lines 113-116.
Line 287. The difference between time of stopping heating and starting heating is not "heating efficiency", but it is "heating time". Please make the corrections in whole text consistently.
Tables 3 and 4. I haven't found the formulas for P-value.
Author Response
Thank you for your attention again. The formulas and descriptions in the new paper have been modified. Detailed changes can be found in the documentation.
